# A modelled evaluation of the impact of COVID-19 on breast, bowel, and cervical cancer screening programmes in Australia

Carolyn Nickson[1,2]*, Megan A Smith[1], Eleonora Feletto[1], Louiza S Velentzis[1,2]*, Kate Broun[3], Sabine Deij[1,2], Paul Grogan[1], Michaela Hall[1], Emily He[1], D James St John[3,4], Jie-Bin Lew[1], Pietro Procopio[1,2], Kate T Simms[1], Joachim Worthington[1], G Bruce Mann[5,6], Karen Canfell[1]

[1]Daffodil Centre, The University of Sydney, a Joint Venture with Cancer Council NSW, Sydney, Australia; [2]Melbourne School of Population and Global Health, University of Melbourne, Melbourne, Australia; [3]Prevention Division, Cancer Council Victoria, Melbourne, Australia; [4]Department of Medicine, The Royal Melbourne Hospital, University of Melbourne, Melbourne, Australia; [5]Breast Service, The Royal Women's and Royal Melbourne Hospital, Melbourne, Australia; [6]Department of Surgery, University of Melbourne, Melbourne, Australia

*For correspondence:
Carolyn.Nickson@nswcc.org.au
(CN);
louizav@nswcc.org.au (LSV)

**Abstract** Australia introduced COVID-19 infection prevention and control measures in early 2020. To help prepare health services, the Australian Government Department of Health commissioned a modelled evaluation of the impact of disruptions to population breast, bowel, and cervical cancer screening programmes on cancer outcomes and cancer services. We used the *Policy1* modelling platforms to predict outcomes for potential disruptions to cancer screening participation, covering periods of 3, 6, 9, and 12 mo. We estimated missed screens, clinical outcomes (cancer incidence, tumour staging), and various diagnostic service impacts. We found that a 12-mo screening disruption would reduce breast cancer diagnoses (9.3% population-level reduction over 2020–2021) and colorectal cancer (up to 12.1% reduction over 2020–21), and increase cervical cancer diagnoses (up to 3.6% over 2020–2022), with upstaging expected for these cancer types (2, 1.4, and 6.8% for breast, cervical, and colorectal cancers, respectively). Findings for 6–12-mo disruption scenarios illustrate that maintaining screening participation is critical to preventing an increase in the burden of cancer at a population level. We provide programme-specific insights into which outcomes are expected to change, when changes are likely to become apparent, and likely downstream impacts. This evaluation provided evidence to guide decision-making for screening programmes and emphasises the ongoing benefits of maintaining screening in the face of potential future disruptions.

## Editor's evaluation

This study presents important results on predicted impact of cancer screening disruptions in Australia during the COVID-19 pandemic based on rapid-response consultation with public health stakeholders. The evidence presented is solid as simulations were based on several previously validated breast, cervical, and bowel cancer screening decision models. Although the scenarios were based on hypothetical disruptions that do not always match experienced disruptions, the work will be of interest to local policymakers, public health specialists, and cancer epidemiologists.

## Introduction

Population cancer screening programmes aim to reduce cancer-specific mortality, complications, and side effects associated with the treatment of advanced-stage neoplasms and, in the case of bowel and cervical screening, reduce the incidence, illness, and mortality of cancers through the detection and treatment of precancerous abnormalities (*Lauby-Secretan et al., 2015*; *Bouvard et al., 2021*; *Lauby-Secretan et al., 2018*). Breast, bowel and cervical screening programmes reduce cancer-specific deaths, with mortality benefits outweighing the risks associated with screening (*Lauby-Secretan et al., 2015*; *Bouvard et al., 2021*; *Lauby-Secretan et al., 2018*).

Australia established national screening programmes for breast and cervical cancer in 1991 and colorectal cancer in 2006 (*Australian Institute of Health and Welfare, 2020a*; *Australian Institute of Health and Welfare, 2021c*; *Australian Institute of Health and Welfare, 2021a*). Following the World Health Organization's declaration of the COVID-19 pandemic, in March 2020, Australia introduced strict national and state-level infection control measures, including physical and social restrictions affecting health services, as well as business, transport, and public gatherings. These restrictions proved highly effective for short-term national pandemic control, although significant outbreaks followed through returning travellers and hotel quarantine programmes (*Victorian Government, 2021*), subsequent new COVID variants (*Australian Institute of Health and Welfare, 2021b*), and ongoing variation in control measures (*Sombetzki et al., 2021*). It was not known to what extent restrictions would impact cancer screening participation, nor the implications of pauses in delivering screening programmes.

Modelled evaluations of COVID-impact scenarios have been undertaken by different countries to support their cancer screening programmes as they respond to the pandemic (*Maringe et al., 2020*; *Burger et al., 2021*; *de Jonge et al., 2021*). In parallel, collaborations within the global modelling community have been established, such as the COVID-19 and Cancer Global Modelling Consortium, to collectively conduct evaluations to support cancer control during and after the COVID-19 pandemic (*COVID-19 and Cancer Global Modelling Consortium, 2022*). In early 2020, we were commissioned by the Australian Government Department of Health to undertake a rapid-response modelled evaluation of the health impacts of potential widespread disruptions to population breast, bowel, and cervical screening. This was the first time such an analysis across all screening programmes had been performed.

This article presents a within-country comparison of key findings, at a national level, between three established population cancer screening programmes, using comparable time periods and metrics. Although some of the scenarios explored here have been harnessed in prior cross-country evaluation studies (*de Jonge et al., 2021*; *Smith et al., 2021*), modelled scenarios in this article were developed after consultation with the Australian Government Department of Health and input from stakeholders to ensure they were relevant to the local context. We demonstrate what could be estimated by a rapid response evaluation based on information available early in the pandemic and discuss how these estimates relate to subsequent observed disruptions to screening. In future, our modelled predictions can be compared to emergent observed national stage and mortality data.

## Methods

We used simulation modelling to estimate outcomes for various potential disruptions to cancer screening services. Modelled scenarios were developed after consultation with the Australian Government Department of Health and input from administrators of the national screening programmes and were designed to be within the capabilities of our comprehensive and validated *Policy1* modelling platform (https://www.policy1.org/) (*Hall et al., 2019*; *Lew et al., 2017*). Scenarios, reported outcomes, and calendar periods varied between screening programmes due to differences in cancer natural histories, existing screening programme protocols, and stakeholder requests and advice.

### Modelling platform

To simulate scenarios and estimate projected outcomes, we used *Policy1* which has evaluated the clinical benefits and harms of a wide range of screening protocols, new technologies, and risk-based approaches aiming to optimise population cancer screening programmes (*Canfell, 2023*).

**Table 1.** Scenarios modelled for each of the three screening programmes and key outcomes reported.
Each scenario is compared to status quo outcomes, which incorporate various population projections based on pre-COVID observed data.

| Screening programme | Disruption scenario by period | | | | Outcomes | | |
| --- | --- | --- | --- | --- | --- | --- | --- |
| | 12 **mo** | 9 **mo** | 6 **mo** | 3 **mo** | Cancer diagnoses | Delayed diagnoses/cancer staging | Screening episodes |
| Breast screening (females, 50–74 y)* | 12-mopause assuming gradual recovery over 6 mo, to 50% higher screening capacity than status quo | N/A | 6-mo pause assuming gradual recovery over 6 mo, to status quo screening capacity | 3-mo pause assuming gradual recovery over 6 mo, to status quo screening capacity | Invasive breast cancers (2020–2021) | Interval cancer rates, tumour size, grade and nodal involvement at diagnosis (2020–2021) | Screening episodes (1 April 2020 to 31 March 2021) |
| Bowel screening (persons, 50–74 y)† | 12-mopause | N/A | 6-mo pause | 3-mo pause | CRC cases (2020–2021; 2020–2060) | Adenomas missed or delayed (2020–2021) CRC cases detected at a later stage (2020–2060) | Number of iFOBT screening kits returned (2020–2021) |
| Cervical screening (females, 25–74 y) ‡ | 12-modisruption; assuming a 95% reduction in primary screening attendance | 9-modisruption assuming a 75% reduction in primary screening attendance | 6-mo disruption assuming a 50% reduction in primary screening attendance | N/A | Additional cervical cancers (2020–2022) | Cervical cancers detected at a later stage (2020–2022) | Number of women screening (2020–2022) |

*Breast screening: status quo: biennial mammography for women 50–74 y old; invitation letter sent at age 50 y; 55% participation rate. All scenarios assume no throughput during the pause period then a gradual recovery of screening throughput, with 3- and 6-mo disruption scenarios reach status quo rates by the seventh month after screening resumption, and the 12-mo disruption scenario reaching screening capacity at 50% higher than status quo by the seventh month after screening resumption. Breast screening outcomes are reported for women aged 50–74 at any time during the reporting period. Two-year reporting periods align with routine reporting of BreastScreen outcomes due to the programme being mostly biennial. Breast programme recall rates describe mammographic screening episodes referred for further assessment, and false-positive recall rates describe recalled screens with a benign final outcome after assessment. Screening programme sensitivity describes the screen-detected cancers as a proportion of screen-detected cancers and interval cancers.

†Bowel screening: status quo: biennial iFOBT screening of men and women 50–74 y old; invitation letter with test sent at age 50 y; participation rate 43.5%. No throughput is assumed during the pause period. Bowel screening outcomes are reported for individuals age-eligible (aged 50–74 y) for screening in either 2020 or 2021, i.e., individuals born in 1945–1971, with reporting periods selected to indicate both the immediate effect and the lifetime effect of the eligible cohort. Outcomes describe the expected number of incident colorectal cancers in the screening cohort only.

‡Cervical screening: status quo: 5-yearly HPV screening of women 25–74 y old; invitation letter sent at age 25 y; participation rate at 46%. Primary screening tests describe women attending for a primary screening test. Scenarios assume women who miss screening in 2020, instead attend over 2021–2022, no disruption to surveillance or colposcopy visits, and no changes to rates of women presenting with symptoms. Cervical screening outcomes are reported for women age-eligible (aged 25–74 y) for screening at any time during 2020–2022, reporting the period 2020–2022 because all women who missed screening in 2020 were assumed to re-attend by 2022, and because 2022 aligns with the last year in the first 5-y-round of primary HPV screening.

CRC: colorectal cancer; iFOBT: immunochemical faecal occult blood test.

## Scenario selection

Scenarios were specified according to screening disruption duration (broadly 3, 6, 9, or 12 mo) and degree (complete pauses or reduced throughput) (*Table 1*).

## Breast screening

The BreastScreen Australia (BSA) programme offers biennial (and some annual) mammographic screening to women from the age of 40 y, targeted to women aged 50–74 y (*Australian Institute of Health and Welfare, 2020a*). The programme is administered by the eight state and territory governments under a national agreement. All abnormal screens are assessed within the programme. Prior to COVID-19, participation by women aged 50–74 was approximately 55%, with 51% of national breast cancers detected through BreastScreen, while the proportion of clients who attended screening on schedule ranged from 60% to 84%, depending on the screening round (*Australian Institute of Health and Welfare, 2020a*). Using the Policy1-Breast model, we evaluated a 3-, 6-, or 12-mo national-level pause (applied as an averaging of annual rates), commencing 1 April 2020. We assumed a gradual restoration of service following pause periods, returning by the seventh month to pre-COVID throughput (screening capacity) for shorter-term pauses (3- and 6-mo) and 150% of pre-COVID throughput for a longer pause (12 mo), on the basis that only a year-long pause would involve expanded recovery strategies. Opportunistic or risk-based screening outside the programme was assumed to be reduced by 50% during programme pause (due to COVID-19 social distancing requirements), reaching 100% on BreastScreen service resumption. For this evaluation, Policy1-Breast was revised to include a prioritisation module (see details in *Supplementary file 1A*) so that it could be applied to capacity-driven recovery scenarios. Following service resumption, the restart strategy

involved the allocation of available screens according to (i) whether clients were newly invited or existing, (ii) their age, (iii) the period in which their appointment fell (i.e. during the pause or during the recovery period), and (iv) the time elapsed since their originally scheduled appointment. Appointments were prioritised for women in the target age range of 50–74 y and to clients most overdue for screening. With this revision, individual clients within the simulation could be channelled into available screening appointments prioritised according to different factors, such as age or screening round.

Additional information on the Policy1-Breast model can be found at https://www.policy1.org/models/breast.

## Bowel screening

The National Bowel Cancer Screening Program (NBCSP) provides biennial primary screening, using an immunochemical faecal occult blood test (iFOBT), to all Australians aged 50–74 y via the national postal service. The screening kit is self-administered at home and, if positive, a follow-up colonoscopy is recommended for diagnostic investigation. Prior to COVID-19, participation was approximately 44%, with 62% of individuals with a positive iFOBT reported as completing a colonoscopy (*Australian Institute of Health and Welfare, 2021c*). Using the Policy1-Bowel model, we evaluated a 3-, 6-, or 12-mo national-level pause, commencing 1 April 2020 during which time we assumed that no screening invitations or kits were sent or processed and all primary screening and diagnostic or surveillance colonoscopies associated with the NBCSP were halted. For modelling purposes, these periods were assumed to occur in 2020 – starting from April for 3- or 6-mo pauses, and over the course of 2020 for the 12-mo pause. It was assumed that individuals missing a screen during the pause would be screened at the next screening round, 2 y later. Individuals with undetected polyps or colorectal cancers (CRC) who missed screening due to the pauses would therefore have these detected in a later year, either symptomatically or at a later screening round. These may be detected as more advanced disease.

The status quo scenario is modelled based on no disruption to the NBCSP using observed participation rates (~40%), with all rates from 2017 onwards extrapolated from the reported data for 2017, unless otherwise noted. Based on the observed data, the model assumes ~70% of individuals with a positive iFOBT complete a diagnostic follow-up colonoscopy. Depending on the follow-up colonoscopy findings, individuals are referred to either return to the NBCSP for iFOBT screening after an interval of 4 y or to repeat colonoscopic assessments (referred to as surveillance colonoscopies) after 1–5 y based on the 2011 guidelines recommendations. Combined, both follow-up and surveillance colonoscopies are referred to as NBCSP-related colonoscopies.

Outcomes have been estimated for the affected cohorts only, that is, among people aged 50–74 y eligible for screening in 2020 and 2021. This cohort comprises 7.1 million people; 3.5 million men and 3.6 million women.

Based on the GESA recommendations from 24 March 2020, most 'elective' colonoscopies were suspended for most of March and April 2020 unless considered to be 'urgent'. Recommendations suggested that colonoscopies for the investigation of a positive iFOBT be considered on a case-by-case basis only if the patient has not had a high-quality colonoscopy within the previous 4 y. The continuation of urgent colonoscopies was not incorporated in these results, nor was rescheduling of elective colonoscopies. We note that recommendations to recommence colonoscopy services were announced at the end of the April 2020 but have not been incorporated into these results.

Additional information on the Policy1-Bowel model can be found at https://www.policy1.org/models/bowel.

## Cervical screening

The National Cervical Screening Program (NSCP) is provided to women aged 25–74 y and usually involves a visit to primary care. In mid-March 2020, the NCSP was approximately 27 mo into a transition from 2-yearly Pap testing to 5-yearly primary HPV screening, where a woman's first HPV test was due 2 y after her previous Pap test.

As part of this evaluation, we updated modelled screening participation to reflect National Cancer Screening Register data (*Supplementary file 1B*), indicating that 53.6% of women had already attended for their first HPV test, and therefore were not due to attend for routine screening again until at least December 2022. We evaluated 6-, 9-, or 12-mo disruptions to screening participation in

2020, with participation during these periods being 50, 75, and 90% lower, respectively, than would otherwise have been expected. We explored two recovery scenarios, one where women who missed screening in 2020 all attended in 2021 (used with the 9-mo disruption scenario), and another where women who missed screening in 2020 attended gradually over 2021–2022 (used with the 6- and 12-mo disruption scenarios).

All modelled scenarios assume the NCSP transitioned from the pre-renewed NCSP (2-yearly cytology for women aged 18–69) to the renewed programme (5-yearly primary HPV screening for women aged 25 to 70–74 y) at the beginning of 2018, including clinical management guidelines with women with detected abnormalities (*Cancer Council Australia, 2017*). The model incorporated recent policy changes that were expected to reduce the impact of COVID-19 disruptions, utilising unpublished NCSR data on the number of women who had received at least one primary HPV screening test and therefore were likely to be unaffected by the disruption (because they are not due for routine screening in 2020 or are already under surveillance). This meant that fewer women were expected to attend for a routine primary HPV test in Australia in 2020 than in 2019 or earlier years, and that women who would have attended in 2020 under the status quo were already overdue for screening (since it had been more than 2 y since their last Pap test), and as a result, a higher risk group.

Similar re-screening patterns as in the pre-renewed NCSP were assumed to apply until women have attended for their first HPV test in the renewed NCSP (i.e. screening behaviour reflects that women are recommended to attend for their first HPV test 2 y after their last cytology test). These screening patterns for women's first HPV test differ slightly from assumed adherence to the 2-yearly interval in the pre-renewed NCSP in order to directly reflect NCSR data on observed behaviour from December 2017 onwards (see data sources for Policy1-cervix model at https://www.policy1.org/models/cervix). Re-screening patterns reflecting a recommended 5-yearly interval do not apply to women until after they have attended for their first HPV test. Assumptions for re-screening attendance (routine testing) and follow-up under all modelled scenarios are outlined in *Supplementary file 1C*.

To calculate how many cancers were upstaged due to the disruption, we assumed that additional cancers that were diagnosed over 2021–2022 were diagnosed at the localised stage, and that any increase in the number of cancers diagnosed at the distant stage was due to cancers being upstaged

**Table 2.** Modelled screening episodes (number of screens and as a proportion of status quo screens) provided under each scenario over the period 1 April 2020–31 March 2021 for people in the target age range (eligible for screening during the affected period) and sex.

| Programme (target sex and age range) | Status quo (comparator) | Disruption scenario by period | | | | Observed number of tests as a proportion of status quo, Jan–Sept 2020 (by quarter and for whole period) (Q1, Q2, Q3, Q1–Q3) |
| | | 12 mo | 9 mo | 6 mo | 3 mo | |
| Breast (females, 50–74) | 973,019 (100%) | 0 (0%) | N/A | 298,113 (31%) | 518,680 (53%) | (98%, 44%, 105%, 82%)* |
| Bowel (persons, 50–74) | 1,353,875 (100%) | 0 (0%)† | N/A | 680,259 (50%) | 1,016,915 (75%) | (54%, 92%, 97%, 81%) ‡ |
| Cervical (females, 25–74) § | 1,413,888 (100%) | 386,451 (27%) | 805,537 (57%) | 1,143,510 (81%) | N/A | (55%, 34%, 39%, 43%) ¶ |
| Combined – provided | 3,740,782 (100%) | 386,451 (10%) | N/A | 2,121,882 (57%) | N/A | |
| Combined – missed | 0 (0%) | 3,250,432 (90%) | N/A | 1,612,032 (43%) | N/A | |

*Percentages derived from a comparison of screens for 2020 vs. 2018, for women aged 50–74 y (AIHW Table 1.2) (*Australian Institute of Health and Welfare, 2020b*).

†No screening provided during the reported period.

‡Percentages derived from a comparison of the number of kits returned for adults aged 50–74 y in 2020 vs. 2019 (AIHW Table 3.2) (*Australian Institute of Health and Welfare, 2020b*).

§Includes women attending for surveillance or other tests, not only routine primary screening tests. For this reason, the percentages do not correspond to the 25–95% reductions assumed among women attending for a routine cervical screening test.

¶Percentages derived from a comparison of the number of HPV tests conducted through the NCSP for women aged 25–74 y, 2020 vs. 2019 (AIHW Table 2.2) (*Australian Institute of Health and Welfare, 2020b*).

from regional to distant. Other changes in the numbers of localised or regional cancers were assumed to be a result of upstaging from localised to regional.

## Reported outcomes

Outcomes reported from each modelled evaluation are shown in *Table 1*. Mortality estimates are not reported; however, estimates for some of the scenarios have since been published separately for cervical and bowel screening (*de Jonge et al., 2021*; *Smith et al., 2021*).

## Results

### Missed screens

For the scenarios modelled, the 3.7 million combined breast, bowel, and cervical screening episodes expected under status quo in the 12 mo starting 1 April 2020 would be reduced by 90% with a 12-mo disruption and by 43% with a 6-mo disruption (*Table 2*).

### Breast cancer screening

#### Population cancer outcomes

For the 2-y period 2020–2021, we estimate that screening pauses of any duration would lead to an overall reduction in population-level invasive breast cancer diagnosis for women aged 50–74 y, noting that the estimated reduction in cancer incidence was comparable (~9%) between a 6-mo pause with pre-COVID throughout during recovery and a 12-mo pause with 150% throughput during recovery (*Table 3*).

All scenarios would lead to reduced population-level rates of screen-detected cancers and interval cancers, reduced programme sensitivity, and more advanced tumour stage at diagnosis. Longer-term fluctuations in clinical and programme outcomes are also expected; for example, with a 12-mo pause we estimate a 10% difference in population-level invasive breast cancer diagnoses among women aged 50–74 between 2020 and 2021 (270 per 100,000 women) and 2022–2023 (296 per 100,000 women) (*Supplementary file 1D*).

### BreastScreen programme outcomes

We assumed higher priority for women who missed scheduled screens during the pause and report the estimated distribution of these women compared to other client groups (*Supplementary file 1E*). Estimated screening intervals ranged from a median of 107 wk for a 3-mo pause through to 130 wk for a 12-mo pause, with a median of 154 wk for women who missed screens during the pause (*Supplementary file 1E*). Recall rates are predicted to fluctuate over time under various pause scenarios, ranging from 5.3% (3-mo pause) to 5.6% (12-mo pause) (*Table 3*), most likely due to an increasing proportion of first-round screening during recovery.

### Bowel screening

#### Population cancer outcomes

All three scenarios illustrated a decrease in cancer diagnoses and cancers being diagnosed at a later stage in the screening cohort compared to the status quo. Across the modelled scenarios, colorectal cancer diagnoses decreased in 2020–2021 by 583–2549 cases (2.8–12.1% decrease). The modelled disruptions resulted in a stage shift, with up to 891 cases which would have been diagnosed but instead progressed to a later stage with lower survival.

Due to the screening disruptions, fewer positive iFOBT would lead to fewer diagnostic follow-up colonoscopies, and additionally no surveillance colonoscopies would be conducted. The NBCSP-related colonoscopies that would not be conducted in 2020 due to the 3-, 6-, and 12-mo disruptions would be 19,151, 38,335, and 76,125 respectively.

This analysis found that a 12-mo pause to screening would lead to an additional 7140 colorectal cancer cases over 2020–2050. This is due to both changes in detection of both cancers and precancerous lesions during the 2020–2021 period, as well as the long-term effect on changes to screening participation and behaviours estimated to be attributable to the disruption.

Additional outcomes for diagnostic assessments and short-term adenoma outcomes are presented in *Supplementary file 1F*.

**Table 3.** Selected estimated outcomes (2020–2021) for the evaluation of disruptions to breast, bowel, and cervical cancer screening. Proportional changes compared to status quo are shown in brackets.

| Outcome | Disruption scenario by period | | | | |
|---|---|---|---|---|---|
| **Breast screening*** | Status quo | 12 mo | 9 mo | 6 mo | 3 mo |
| Invasive breast cancers per 100,000 women[†] | 298 | 270 (–9%) | NA | 272 (–9%) | 286 (–4%) |
| Screen-detected invasive breast cancers per 100,000 women[†] | 127 | 97 (–24%) | NA | 99 (–22%) | 117 (–8%) |
| Interval cancers (12 mo) [‡] | 15 | 8 (–47%) | NA | 11 (–27%) | 13 (–13%) |
| Interval cancers (27 mo) [‡] | 38 | 33 (–13%) | NA | 33 (–13%) | 35 (–8%) |
| Programme sensitivity [§] | 76.8% | 74.8% (–3%) | NA | 75.2% (–2%) | 77.0% (0%) |
| Tumour size (% ≤15 mm diameter) | 59.7% | 56.5% (–5%) | NA | 58.3% (–2%) | 59.6% (0%) |
| Nodal involvement (% involving nodes) | 24.9% | 26.4% (6%) | NA | 25.1% (1%) | 25.0% (0%) |
| Grade (% grade 3 versus grade 1/2) | 46.6% | 48.4% (2%) | NA | 47.2% (1%) | 46.7% (0%) |
| Recall rate (N) [¶] | 5.2% | 5.6% (8%) | NA | 5.3% (2%) | 5.3% (2%) |
| False-positive recall rate (N)[**] | 4.6% | 4.8% (4%) | NA | 4.6% (0%) | 4.7% (2%) |
| **Bowel screening** | | | | | |
| Colorectal cancer diagnoses | 21,068 | 18,518 | NA | 19,844 | 20,484 |
| Change (%) in 2020–2021 | - | –2549 (–12.1%) | NA | –1223 (–5.8%) | –583 (–2.8%) |
| Undetected cancers which would advance in stage in 2020–2021 | - | 891 | NA | 529 | 261 |
| % of cancers detected at stages 3–4, 2020/2021 | 33.9% | 40.7% (7%) | NA | 35.9% (2%) | 34.6% (1%) |
| Colonoscopies in 2020–2021 | 194,954 | 118,829 | NA | 156,619 | 175,804 |
| Change (%) in 2020–2021 | - | –76,125 (–39.0%) | NA | –38,335 (–19.7%) | –19,151 (–9.8%) |
| **Cervix screening[††, ‡‡]** | | | | | |
| Cervical cancer diagnoses | 1878 | 1947 | 1912 | 1899 | NA |
| Increase (%) | | 69 (3.6%) | 34 (1.8%) | 21 (1.1%) | NA |
| Upstaged cancers | | | | | |
| Localised → regional | | 18 | 8 | 6 | NA |
| Regional → distant | | 9 | 4 | 3 | NA |
| % cancers upstaged [§§] | | 1.4% | 0.6% | 0.5% | NA |
| Colposcopies | 245,620 | 211,445 | 230,383 | 233,463 | NA |

*Breast screening: for all scenarios, screening is assumed to resume gradually after the pause to services, reaching status quo rates by the seventh month after resumption for the 3 and 6 mo scenarios.

[†]Breast screening: rates are per 100,000 women in the Australian population, including women who do not usually participate in screening.

[‡]Breast screening: invasive breast cancers arising within 12 or 27 mo of a negative screening episode. Figures reflect interval cancers diagnosed in 2020, or 2020–2021, respectively.

[§]Breast screening: screen-detected cancers as a proportion of screen-detected + interval cancers (27 mo).

[¶]Breast screening: the proportion of screening episodes recalled for further investigation.

[**] Breast screening: the proportion of screening episodes recalled for further investigation, with a benign final outcome after that investigation.

[††]Cervical screening: the 12-, 9-, and 6 mo scenarios assume a decrease in attendance of 95, 75, and 50%, respectively, compared to what would otherwise have been expected in 2020.

[‡‡]Cervical screening: All values are rounded to whole numbers.

[§§]Number of upstaged cancers (localised to regional or regional to distant) as a percentage of the number of cancer cases predicted under the status quo scenario.

## Cervical screening

### Population cancer outcomes

All three scenarios resulted in an increase in cancer diagnoses and cancers being diagnosed at a later stage among screening-age women compared to what would have been expected in the absence of any disruption. The increase in cancer cases over 2020–2022 ranged from 21 to 69 cases (1.1–3.6% increase). Most (57–63%) of the additional cancers were diagnosed among women 30–49 y. Women aged 30–39 y and 40–49 y were also the age groups where the percentage increase in cancers was largest (although still relatively small: 1.1–4.1% and 1.2–4.3% in women aged 30–39 and 40–49 y, respectively). The model predicted that disruptions to routine primary screening would lead to 6–18 cervical cancers being diagnosed at regional stage in 2021–2022, rather than as localised cancers in 2020; and 3–9 cervical cancers being diagnosed at distant stage in 2021–2022, rather than at regional stage in 2020. Considering both additional cervical cancers and those which were diagnosed at a later stage due to disruptions to cervical screening, an estimated 30–97 women would be affected by delays in diagnosis due to disruptions to routine screening in 2020. The longer-term impact of these additional and upstaged cancers on cervical cancer deaths is presented in *Supplementary file 1G*, and the impact of disruptions on the number of women expected to attend a cervical screening test is presented in *Supplementary file 1H*.

### Colposcopy demand

Disruptions to primary screening in 2020 are estimated to result in 17,680–47,868 fewer women attending for a colposcopy in 2020 (17.4–47.2% reduction) as a result of their primary screening or triage test. The impact of the disruptions on expected colposcopy utilisation between 2020 and 2022 is presented in *Supplementary file 1I*.

### Additional results

Additional results can also be found in the online reports produced as part of this project for the Australian Government, Department of Health (*Australian Government, Department of Health, 2020c*; *Australian Government, Department of Health, 2020d*; *Australian Government, Department of Health, 2020a*; *Australian Government, Department of Health, 2020b*).

## Discussion

We present outcomes estimated in 2020 for a range of potential disruptions to national population cancer screening programmes for breast, cervical, and colorectal cancer, including population-level changes in cancer diagnoses and staging. For similar disruption scenarios, we estimate markedly different impacts for each programme. We estimated that a 12-mo screening disruption from early 2020 would in the short term lead to reduced diagnoses of breast cancer (9.3% population-level reduction over 2020–2021, 150% screening throughput) and colorectal cancer (up to 12.1% reduction over 2020–21) and increased cervical cancer diagnoses (up to 3.6% over 2020–2022), with upstaging expected for all three cancer types. For a 6-mo screening disruption we estimated reduced diagnoses of breast cancer (9.4% population-level reduction over 2020–2021, with 100% throughput) and colorectal cancer (up to 5.8% reduction over 2020–21) and increased cervical cancer diagnoses (up to 1.1% over 2020–2022), with less marked upstaging compared to a 12-mo disruption.

Findings were most marked for bowel screening, for which short-term upstaging and reduced cancer incidence was expected to lead to a significant number of additional colorectal cancer cases over the lifetime of the affected cohort. For breast screening, we estimated smaller but important reductions in cancer diagnoses at a population level over 2021 with fluctuating rates in subsequent years and upstaging of cancers at diagnosis (noting that estimated reductions in population-level interval cancer rates are counter-intuitive but an artefact of reduced screening participation, as interval cancers are defined with reference to screening events). For cervical screening, relative and absolute increases in cancer are expected to be less than for breast and colorectal cancer. This is likely due to COVID-19 impacts coinciding with lower participation under status quo because the programme recently transitioned to a longer screening interval, along with the lower burden of disease due to prevention through both screening and widespread uptake of HPV vaccination (80% three-dose coverage among females aged 14–15) (*Australian Government, Department of Health, 2022a*).

## Strengths and limitations

We used established modelling platforms for breast, colorectal, and HPV/cervical cancer natural history and screening in Australia, which were already well calibrated and validated to the Australian context using high-quality published data, including detailed governmental screening reports. These models describe the Australian national age and sex distribution and the incidence and natural histories of breast, cervical, and colorectal cancer, and can be used to simulate a range of scenarios for cancer screening. Prior investment into these modelling platforms facilitated the rapid response modelling required in the face of a crisis; the evaluations reported highlight the importance of ongoing investment into comprehensive, well-calibrated and validated population predictive models that can be quickly harnessed for critical decision-making as needed for disruptions such as the COVID-19 pandemic and other potential disruptions due to major social, public health, or natural disasters as are increasingly common due to climate change (*Bureau of Meteorology, 2020*).

This evaluation included detailed reporting of outcomes for different subgroups where this was requested by key stakeholders (e.g. breast screening outcomes according to whether clients were directly or indirectly impacted by screening disruptions).

Each modelled evaluation drew on the most contemporary data available, employing projections as required. In addition, the cervical screening model incorporated the effects of recent policy changes that were expected to reduce the impact of COVID-19 disruptions, drawing on unpublished data from the NCSR.

Scenarios were by necessity defined in simple terms. For example, we modelled single, continuous periods of national disruption but, in practice, screening participation would likely fluctuate by time and place in line with local outbreaks and pandemic control measures. By deliberately focusing on a high-level effect – a wide range of possible effects on attendance for routine screening – the analysis was agnostic to the cause, and therefore this could reflect any or many of a wide range of possible factors, including changes in behaviour and access to screening, reduced capacity of services involved in providing screening, or reduced saliency or delivery delays of reminder letters or home-delivered testing kits. These effects will be incorporated in future modelled evaluations, after careful calibration and validation to observed data, with a view to extending the modelled outcomes to mortality estimates.

Modelled estimates will likely include some biases. We do not account for changes in diagnostic and treatment services due to COVID-19 which have since been observed (*Australian Government, Cancer Australia, 2021*). For example, Medicare Benefits Schedule claims data from 2020 indicates a 4% reduction in breast diagnostic services and 6% reduction in breast cancer-related surgical procedures, and for colorectal cancer a 13% reduction in diagnostic services (mostly colonoscopies) and 1% reduced surgical procedures. For cervical cancer screening, estimates of additional and upstaged cervical cancer diagnoses potentially underestimate the overall impact of the pandemic on cervical cancer diagnoses.

## Subsequent observed outcomes

This modelled evaluation was conceived during the first wave of COVID-19 restrictions, and in the context of needing to provide safe health services including population screening for breast, cervical, and colorectal cancer. At the time of the evaluation, the current analysis provided government with timely background estimates of potential impacts. The Australian cervical and bowel screening programmes continued to operate continuously through 2020 (and since), while BreastScreen services paused nationally from March to late April/early May 2020, resuming thereafter. Monthly screening tests over the period January–September 2020 fluctuated as shown in *Figure 1* (and *Figure 2*), (*Australian Institute of Health and Welfare, 2020b*) with an overall 34% reduction in screening tests across the three programmes comprising over 1 million (1,035,710) fewer breast, bowel, and cervix tests in age-targeted Australian populations compared to January to September 2019 (*Australian Institute of Health and Welfare, 2020b*). For the year 2020, observed reductions in participation were lower than the most conservative estimates modelled for each programme, with 14% fewer breast screening tests (*Australian Institute of Health and Welfare, 2023*), an estimated 7% reduction in bowel screening participation (*Australian Institute of Health and Welfare, 2023*) and for cervical screening, based on reimbursement claims data, a 10% reduction in expected HPV tests (*Smith et al., 2016*; *Australian Government, Services Australia, 2023*).

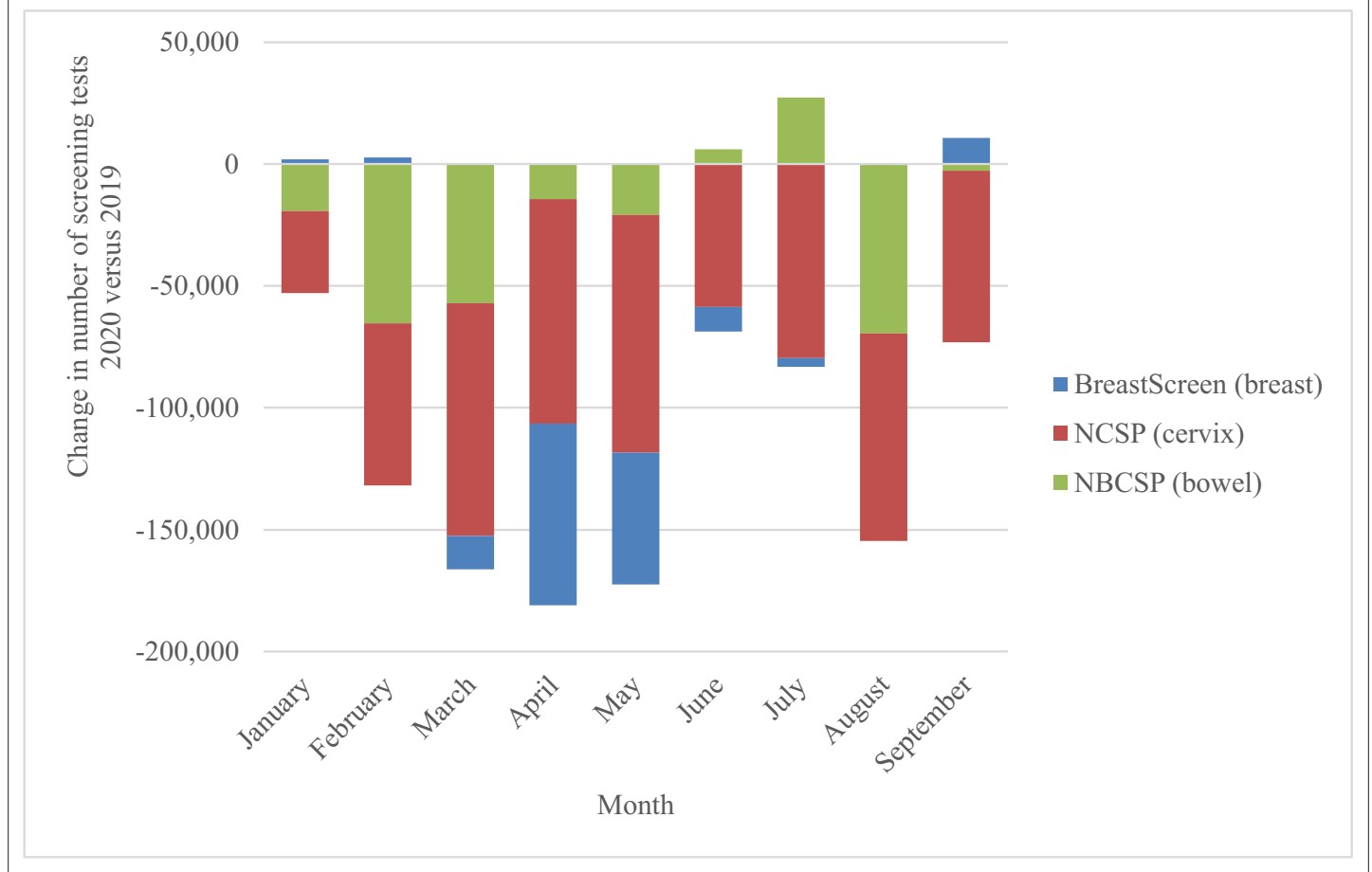

**Figure 1.** National change in number of screening tests by month, 2020 vs. 2019, for the breast, bowel, and cervical screening programmes. Derived from AIHW 2020 report – Cancer Screening and COVID19 in Australia. *Australian Institute of Health and Welfare, 2020b*. Note: cervical screening test volumes were anticipated to be lower in 2020 than in 2019 due to the extension from a 2-y to a 5-y screening interval (*Smith et al., 2016*).

Observed screening reductions were driven by a range of factors such as screening service closure followed by reduced throughput capacity for breast screening, reduced primary care visits (cervical screening), as well individuals likely de-prioritising screening in the context of the pandemic (*Bittleston et al., 2022*). Screening test rates varied markedly between states and territories during the pandemic, (*Australian Institute of Health and Welfare, 2020b*) consistent with localised, often state-and-territory based controls, including stay-at-home orders, border closures, quarantine programmes, and other physical and social distancing measures. Our evaluation of a range of scenarios indicates the expected nature, scale, and time frame of key outcomes for each screening programme, providing valuable insights relevant to both national and more localised disruptions and highlighting the importance of maintaining screening participation rates.

## Future insights

Our predicted clinical implications of reduced participation cannot be confirmed for some time; however, early signs suggest an observed reduction in cancer diagnosis in 2020 (*Victorian Cancer Registry. Cancer in Victoria, 2020*). While COVID-19 vaccination coverage is high in Australia (two dose: >95% nationally for people >16 y old; three dose: 71%>16 y) (*Australian Government, Department of Health, 2022b*) social distancing measures are a continuing requirement and the redeployment of health services may be required in future if new variants of concern with higher transmissibility and/or virulency take over. As observed data on COVID-impacted screening behaviour for specific sub-populations becomes available, further modelling can help estimate the impact of screening disruptions for different population sub-groups, noting that Australian screening data is regularly

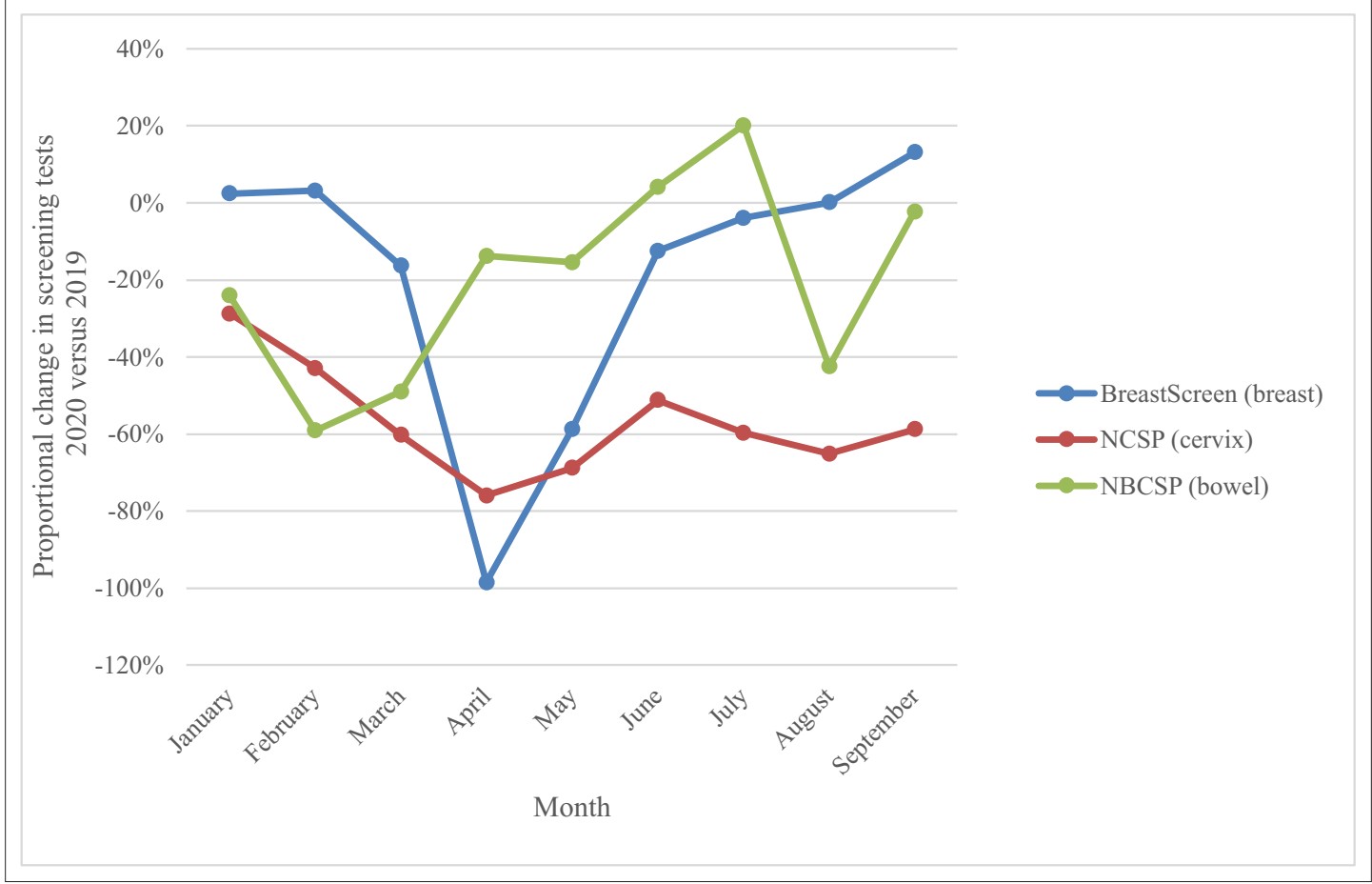

**Figure 2.** National proportional change in number of screening tests by month, 2020 vs. 2019, for the breast, bowel, and cervix screening programmes. Derived from AIHW 2020 report – Cancer Screening and COVID19 in Australia. *Australian Institute of Health and Welfare, 2020b*. Note: cervical screening test volumes were anticipated to be lower in 2020 than in 2019 due to the extension from a 2-y to a 5-y screening interval (*Smith et al., 2016*).

reported by age group, sex (for bowel cancer screening), and various socio-economic indicators. Each screening programme is subject to detailed monitoring and evaluation, working to a range of performance indicators. These indicators are expected to be impacted by service disruptions, in a variety of ways. For example, breast screening interval cancer rates (a key BreastScreen performance indicator) are predicted to vary; this is logical given that the likelihood of an interval cancer is expected to depend on the time since the screen prior to the interval cancer reference screen, screening round, and age, all of which vary for the scenarios evaluated. These changes would impact usual quality assurance monitoring of the programme and, potentially, community perception of programme performance. Therefore, one of the implications of the current evaluation is to focus future strategies for communication around the importance of screening programmes and the need for ongoing high participation as they play a key role in maintaining community confidence in screening.

We quantified resource requirements for programmes and related health services in the face of disruption, with requirements for breast cancer screening assessments, colonoscopies, and colposcopies affected to varying degrees. Variations in breast and colorectal cancer incidence are expected to have a significant flow-on effect on the demand for treatment services, compounded by a changing case-mix with shifts to later-stage diagnoses to varying degrees over time.

## Implications

Collectively, these findings from the 6–12-mo disruption scenarios illustrate that maintaining screening participation is critical to preventing an increase in the burden of cancer at a population level. We enumerate the extent to which COVID-19 disruptions are likely to impact short- and long-term cancer

outcomes, resource requirements, and usual indicators used in programme monitoring and evaluation, providing critical evidence to guide cancer screening programmes as they continue to adapt to the ongoing impacts of the COVID-19 pandemic, and as they prepare for other major disruptions that may arise in the future (*Victorian Cancer Registry. Cancer in Victoria, 2020*).

The estimated impacts described here may be reduced if screening programmes sustain or revisit planned improvements that commenced prior to the pandemic, some of which may also directly assist programme recovery. For example, from 1 July 2022, all women now have the option to use self-collection and this could facilitate continuation or recovery of cervical screening programmes (*Australian Government, Department of Health Ministers, 2021*).

As programmes work to bring screening participants back on schedule, well-planned reactive risk-based approaches may help direct limited services to those who will benefit the most. For HPV screening, this could mean more refinements to triaging and surveillance of screen-positive women, and optimising screening for women protected by HPV vaccination. For breast screening, this could involve prioritising women at higher risk of breast cancer, such as women usually offered annual screening and women assessed as higher risk using routine risk prediction models incorporating breast density (*Harkness et al., 2020*). For bowel screening, risk-stratified prioritisation of people could be rapidly implemented by modifying the faecal occult blood threshold and extending the period of time over which people are asked to complete their missed screens (*van Wifferen et al., 2022*). Future improvements could also consider starting screening at an earlier age for Aboriginal and Torres Strait Islander peoples who are at higher risk of colorectal cancer (*Lew et al., 2022*). For all screening programmes, individuals who missed screening during the disruption should be encouraged to return to screening as soon as it is safe to do so. For bowel screening, previously published modelling estimated that this 'catch-up' screening could almost ameliorate the impact of a disruption (*de Jonge et al., 2021*). As such, one implication of the current evaluation is that screening programmes have the potential to continue to improve outcomes and therefore 'build back better' in the future.

## Acknowledgements

Australian Government Department of Health commissioned Cancer Council NSW (HEALTH/19-20/PH20/13982 C) to develop and conduct the modelled evaluations reported in this manuscript.

## Additional information

### Competing interests

Megan A Smith: reports salary support via fellowship grants from the NHMRC of Australia and Cancer Institute NSW and contracts paid to her institution (the Daffodil Centre) with the Commonwealth Department of Health (Australia) and National Screening Unit (New Zealand). Eleonora Feletto, Jie-Bin Lew: Karen Canfell: is a PI of an investigator-initiated trial of cervical screening, (Compass;ACTRN12613001207707 and NCT02328872) run by the Australian Centre for the Prevention of Cervical Cancer, which is a government-funded not-for-profit charity; the Australian Centre for the Prevention of Cervical Cancer has received equipment and a funding contribution from Roche Molecular Diagnostics. KC is also co-PI on a major investigator-initiated implementation program Elimination of Cervical Cancer in the Western Pacific (ECCWP) which will receive support from the Minderoo Foundation, the Frazer Family Foundation and equipment donations from Cepheid Inc Neither KC nor her institution on her behalf have received direct funding from industry for any project. KC receives salary support from a National Health and Medical Research Council (NHMRC) of Australia fellowship grant. The other authors declare that no competing interests exist.

### Funding

No external funding was received for this work.

### Author contributions

Carolyn Nickson, Conceptualization, Supervision, Methodology, Writing – original draft, Result interpretation; Megan A Smith, Supervision, Methodology, Writing – review and editing, Result

interpretation; Eleonora Feletto, Supervision, Writing – review and editing, Result interpretation; Louiza S Velentzis, Writing – review and editing, Result interpretation; Kate Broun, Writing – review and editing, Result interpretation; Sabine Deij, Formal analysis, Writing – review and editing, Result interpretation; Paul Grogan, Writing – review and editing, Result interpretation; Michaela Hall, Formal analysis, Writing – review and editing, Model refinement; Emily He, Formal analysis, Writing – review and editing; D James St John, Writing – review and editing, Result interpretation; Jie-Bin Lew, Formal analysis, Writing – review and editing, Model refinement; Pietro Procopio, Formal analysis, Writing – review and editing, Model refinement; Kate T Simms, Formal analysis, Writing – review and editing, Model refinement; Joachim Worthington, Formal analysis, Writing – review and editing, Model refinement; G Bruce Mann, Writing – review and editing, Result interpretation; Karen Canfell, Conceptualization, Supervision, Writing – review and editing, Result interpretation

**Author ORCIDs**
Carolyn Nickson (iD) http://orcid.org/0000-0001-5370-6590
Louiza S Velentzis (iD) http://orcid.org/0000-0002-9309-0492
Jie-Bin Lew (iD) http://orcid.org/0000-0003-2837-3565
Joachim Worthington (iD) http://orcid.org/0000-0002-8830-0520

**Decision letter and Author response**
Decision letter https://doi.org/10.7554/eLife.82818.sa1
Author response https://doi.org/10.7554/eLife.82818.sa2

## Additional files

**Supplementary files**
• Supplementary file 1. Suggest 'Additional model specifications and estimates'. (A) Ranking of women aged 50–74 in the queue for available screens during the recovery period. (B) Proportion of eligible women with any Cervical Screening Test recorded – modelled assumption prior to disruption compared to observed data (1 December 2017–14 January 2020), by age. (C) Routine and follow-up attendance assumptions for counterfactual (no-disruption) and disruption scenarios to the NCSP. (D) Selected estimated outcomes by 2-y calendar period (2020–2021), 2021–2022, and 2022–2023 assuming a 12-mo disruption, for women aged 50–74. (E) Estimated screening delays and participant profiles for each BreastScreen programme disruption scenarios modelled, reported for various calendar periods. (F) Diagnostic assessments and short-term adenoma outcomes. (G) Impact of disruption on cancer outcomes among women aged 25–74 y, 2020–2022. (H) Impact of disruption on number of women expected to attend for an HPV test (any purpose). (I) Impact of disruption on expected colposcopy utilisation, 2020–2022.

• MDAR checklist

## Data availability

Supporting information on data sources informing each model can be found on-line (https://www.policy1.org/). The current manuscript is a computational study involving modelling rather than direct analysis of primary datasets and no new datasets have been created.Codes for the Policy1 models have been developed over decades, are proprietary property, and cannot be provided directly by the authors. For access, interested researchers should contact the corresponding author in the first instance. A detailed project proposal would need to be submitted and assessed internally. If successful, appropriately resourced supervision of researchers will be conducted by Daffodil Centre staff.

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
