## [Editor Report]

This study presents important results on predicted impact of cancer screening disruptions in Australia during the COVID-19 pandemic based on rapid-response consultation with public health stakeholders. The evidence presented is solid as simulations were based on several previously validated breast, cervical, and bowel cancer screening decision models. Although the scenarios were based on hypothetical disruptions that do not always match experienced disruptions, the work will be of interest to local policymakers, public health specialists, and cancer epidemiologists.

---

## [Decision Letter]

**Decision letter after peer review:**

Thank you for submitting your article "The impact of COVID-19 on population cancer screening programs in Australia: modelled evaluations for breast, bowel and cervical cancer" for consideration by *eLife*. Your article has been reviewed by 3 peer reviewers, and the evaluation has been overseen by a Reviewing Editor and a Senior Editor. The reviewers have opted to remain anonymous.

Essential revisions:

– Please explain the novelty of this study in relation to previous papers from the same research group.

– Address all requests for clarifications made by reviewers.

– Place figures currently in the Discussion in the Appendix.

– Temper claims regarding the impact of screening disruptions on cancer burden, as the results also suggest that relatively short (~3 month) disruptions would likely not have a large impact on cancer burden.

– Provide some further explanation on why the authors consider that model-estimated reductions in screening tests in the first year of disruption overlapped with observed rates, as Reviewer #1 is correct that empirical disruptions do not appear to always match hypothetical modelled disruptions.

– Provide confidence or uncertainty intervals for model predictions, or provide an explanation as to why confidence intervals are unavailable.

– The definition of interval cancers and findings regarding interval cancers need to be better explained, as the results were confusing and counter-intuitive to reviewers.

– While I understand there are reasons why the authors did not include cancer mortality outcomes as the models did not include the impacts of treatment disruptions, the lack of mortality outcomes was noted by two reviewers so it is likely that many readers would similarly wonder why cancer death outcomes were not included. The authors may want to consider including screening disruption-related predicted additional deaths if available, specifying that these deaths do not include the impact of treatment disruptions.

*Reviewer #1 (Recommendations for the authors):*

Thank you for the nicely written paper. I still have a few comments.

Overall:

1. I recently read two other papers which used the Policy1 modelling platforms to estimate the effect of disruptions in the bowel and cervical screening program on cancer outcomes. I assume that you are familiar with those studies as some of your co-auteurs are also listed on them. I am talking about the following two papers: "Impact of the COVID-19 pandemic on faecal immunochemical test-based colorectal cancer screening programmes in Australia, Canada, and the Netherlands: a comparative modelling study" and "Impact of disruptions and recovery for established cervical screening programs across a range of high-income country program designs, using COVID-19 as an example: A modelled analysis".

Could you please explain the difference between the current study and the two studies I listed above? Why did the current study need to be performed? As you already had the other two studies, which look rather similar to the current study.

2. Make sure you write percentages the same throughout the article. Sometimes I see 9.3 and then I see 9·3 (dot in the middle instead of on the bottom).

Abstract:

1. I miss some information about upstaging. How many percent of the tumours are expected to be upstaged?

2. I believe that a suspension of the screening program for 3 months will not lead to significant upstaging. The first sentence of the conclusion might be a bit too harsh, as suspension for only three months might be wise in situations with limited resources.

Introduction:

1. Nicely written introduction. However, I would like you to mention here the significance of this paper, as there are already two modelling studies using the Policy1 modelling platform.

Methods:

1. I miss a description of the restart strategy of the screening program. Did the persons with a suspended screen still received their screening when the program restarted? Did the program tried to catch-up on the missed screens by increasing their capacity? The current article nicely describes different restart strategies: Kregting, L. M., Kaljouw, S., de Jonge, L., Jansen, E. E., Peterse, E. F., Heijnsdijk, E. A., … and de Kok, I. M. (2021). Effects of cancer screening restart strategies after COVID-19 disruption. British journal of cancer, 124(9), 1516-1523.

2. In line 109 you mention: "60-84% of clients screened on schedule (first to third and subsequent round)". What is meant by this?

3. In line 178 you mention a "cervical screening test (CST)". Do you mean a HPV-test?

Results:

1. Table 3 shows the reduction in the number of screen-detected invasive cancers per 100,000 women. I am quite surprised to see that the number of screen-detected tumors after a disruption of 12 months only decreased by 24%, when there was no screening for an entire year and the screening program restarted gradually over the next 6 months. Is there a catch-up in those last 6 months when the normal screening program was again in place?

2. Table 3 shows the reduction in the number of interval-detected invasive cancers. Does this concern the interval-detected invasive cancers of women screened in 2020/2021? Or does it concern the number of interval-detected invasive cancers diagnosed in 2020-2021?

3. In line 259-260 you write that an estimated 759-3022 people would be affected by delays in diagnosis for bowel cancer. Where can I find those numbers?

Discussion:

1. I was quite surprised to see some figures in the Discussion. I would place figure 1 in the Results section. Also it would be very interesting to add an analysis showing the expected effect of what really happened with the screening program. This would also distinguish your paper from the other two papers I mentioned above.

2. I would place figure 2 in the appendix, since you already have Table 2.

3. Could you please explain your reasoning in line 376-378? I am not sure if I agree with the statement that your model-estimated reductions in screening tests in the first year of disruption overlapped with observed rates. Table 2 show that in Q1-Q3 still 82% of the tests were performed for breast cancer, while with a 3 months disruption only 53% of the test will be performed in April 2020-March 2021. I do not think those numbers overlap. The numbers are also not very comparable as the observed numbers are presented for Jan-Sept 2020 and the modelled numbers for April 2020-March 2021. It would be nice if you can take the same periods.

4. In line 413-414 you mention that your findings illustrate that maintaining screening participation is critical to reducing the burden of cancer at a population level. Could you please explain your reasoning? I believe that the numbers show that screening can safely be suspended for a few months if there are limited resources available. Only if you suspended the program for a year, problems may occur.

*Reviewer #3 (Recommendations for the authors):*

While the presented results by assessing various disrupted screening schedules (12 months, 9 months, 6 months, and 3 months) through statistical modelling are very helpful for guiding policy-makers to avert long-term impact on cause-specific mortality and cost for example in the post-pandemic COVID-19 era, the manuscript can be substantially improved in the light of the following suggestions.

1. It is very important to report the influences of COVID-19 pandemic on long-term mortality resulting from the disrupted screening schedule although the reported outcomes here have already included the reduction of screen-detected cancers and the assessment of upstaging. The author has already indicated the unfeasibility of reporting mortality estimates because of unavailable information on COVID-19 disruptions to cancer treatment. However, since your modelling has assumed many situations on various delayed screening schedule it would not be difficult to model mortality estimates by setting various circumstances on COVID-19 disruptions to cancer treatment.

2. While modelling screening episodes by various disrupted screening schedules plays an important role in policy analysis it is even important to report the empirical results of the delayed screening schedule until 31 March 2021 instead of only reporting the observed number from January to September 2020. You can still report your simulated results for policy analysis but report which scenario, on average, is close to your empirical estimate. So doing is very helpful for predicting the best-case outcomes after 2021.

3. It is quite unreasonable to have seen reduced interval cancers and the reduced program sensitivity with the disruption scenario from the aspect of screening theory. If disruption to screening happen those cancers would be delayed to have diagnosis. It is therefore inadequate to define interval cancer in this manner in the face of disruption scenario, saying an artefact of reduced screening participation. The reader is very confused with such a kind of results. As interval cancer is often a negative outcome in the theory of screening, the results of reducing interval cancer as a result of disruption to screening would be very misleading. The longer disruption the less likely to have interval cancer. This part should be rectified to have adequate presentation.

4. Instead of reporting absolute number of upstaging with different expressions for various cancers, it is indispensable to report relative risk of having upstaging for various disruption scenarios as opposed to status quo by breast cancer, colorectal cancer, and cervical cancer base on the modelling results. So doing this can be also of assistance to evaluate statistical significance by reporting 95% confidence interval and also P value for hypothesis testing.

5. The results of upstaging are so important for this manuscript particularly when mortality estimates have been not reported. However, the modelling results on upstaging seems to vary with disruption scenarios. This part has to be emphasized in the main conclusions for different cancer. Again, using the relative risk estimates would make the comparisons of the finding across three sites of cancer comparable.

6. The point 2 mentioned above is also important for validating the simulated results of screen-detected cases, upstaging and other outcomes with the scenario in commensuration with the empirical scenario of disruption to screening.

---

## [Author Response]

Reviewer #1 (Recommendations for the authors):Thank you for the nicely written paper. I still have a few comments.Overall:1. I recently read two other papers which used the Policy1 modelling platforms to estimate the effect of disruptions in the bowel and cervical screening program on cancer outcomes. I assume that you are familiar with those studies as some of your co-auteurs are also listed on them. I am talking about the following two papers: "Impact of the COVID-19 pandemic on faecal immunochemical test-based colorectal cancer screening programmes in Australia, Canada, and the Netherlands: a comparative modelling study" and "Impact of disruptions and recovery for established cervical screening programs across a range of high-income country program designs, using COVID-19 as an example: A modelled analysis".Could you please explain the difference between the current study and the two studies I listed above? Why did the current study need to be performed? As you already had the other two studies, which look rather similar to the current study.

Modelled scenarios in the current paper (as noted in the first paragraph of the methods section) were developed after consultation with the Australian Government Department of Health and input from stakeholders to ensure they are relevant to the local context. Although some of the scenarios explored in the current paper have been harnessed in prior evaluation studies, as mentioned by the reviewer, they were aimed at exploring how the effect of standardised disruptions scenarios might vary depending on country-specific factors, such as program design. Additionally, the current paper provides a within-country comparison between three established population cancer screening programs, using comparable time periods and metrics to enable comparisons not possible based on previously published findings. Modelled estimates are also presented against subsequent observed national data. We have addressed these points in the last paragraph of the introduction with the following text:

“The current article presents a within-country comparison of key findings, at a national level, between three established population cancer screening programs, using comparable time periods and metrics. Although some of the scenarios explored here have been harnessed in prior cross-country evaluation studies (de Jonge et al., 2021; Smith et al., 2021), modelled scenarios in the current paper were developed after consultation with the Australian Government Department of Health and input from stakeholders to ensure they were relevant to the local context.”

2. Make sure you write percentages the same throughout the article. Sometimes I see 9.3 and then I see 9·3 (dot in the middle instead of on the bottom).

Thank you for pointing that out. This has now been amended.

Abstract:1. I miss some information about upstaging. How many percent of the tumours are expected to be upstaged?

Additional information on upstaging has been added to Table 3 for bowel and cervical screening (please refer to main article). We have also amended the abstract with the text as shown below:

We estimated that a 12-month screening disruption would reduce breast cancer diagnoses (9.3% population-level reduction over 2020-2021) and colorectal cancer (up to 12.1% reduction over 2020-21), and increase cervical cancer diagnoses (up to 3.6% over 2020-2022), with upstaging expected for these cancer types (2%, 1.4% and 6.8% for breast, cervical and colorectal cancers, respectively).

2. I believe that a suspension of the screening program for 3 months will not lead to significant upstaging. The first sentence of the conclusion might be a bit too harsh, as suspension for only three months might be wise in situations with limited resources.

We agree with the reviewer that more significant impacts were predicted for all screening programs when the disruption period was longer than 3 months. We have amended the first line of the Abstract’s conclusion section as shown by the text below:

“Findings for 6-12-month disruption scenarios illustrate that maintaining screening

participation is critical to preventing an increase in the burden of cancer at a population level.”

We would, however, like to point out that for individuals who missed their scheduled bowel screen during the pause, this screen would not be occurring until the next scheduled round. This point has now been added to the methods section under ‘Bowel screening’ as follows:

“Individuals with undetected polyps or colorectal cancers (CRC) who missed screening due to the pauses would therefore have these detected in a later year, either symptomatically or at a later screening round. These may be at detected as more advanced disease.”

Introduction:1. Nicely written introduction. However, I would like you to mention here the significance of this paper, as there are already two modelling studies using the Policy1 modelling platform.

This point has been addressed above (see point 1). The last paragraph of the introduction has been amended to:

“The current article presents a within-country comparison of key findings, at a national level, between three established population cancer screening programs, using comparable time periods and metrics. Although some of the scenarios explored here have been harnessed in prior site-specific cross-country evaluation studies (de Jonge et al., 2021; Smith et al., 2021), modelled scenarios in the current paper were developed after consultation with the Australian Government Department of Health and input from stakeholders to ensure they were relevant to the local context. We demonstrate what could be estimated by a rapid response evaluation based on information available early in the pandemic, and discuss how these estimates relate to subsequent observed disruptions to screening.”

Methods:1. I miss a description of the restart strategy of the screening program. Did the persons with a suspended screen still received their screening when the program restarted? Did the program tried to catch-up on the missed screens by increasing their capacity? The current article nicely describes different restart strategies: Kregting, L. M., Kaljouw, S., de Jonge, L., Jansen, E. E., Peterse, E. F., Heijnsdijk, E. A., … and de Kok, I. M. (2021). Effects of cancer screening restart strategies after COVID-19 disruption. British journal of cancer, 124(9), 1516-1523.

For BreastScreen Australia, the restart strategy is determined by the prioritisation module, described in the Table entitled Supplementary file 1A, where clients are prioritised (ranked) according to (i) whether they are newly invited or existing clients, (ii) their age; (iii) in which period their appointment fell (i.e. during the pause or during the recovery period), and (iv) the time elapsed since their originally scheduled appointment. Clients with the highest priority of being allocated an appointment after the pause, were those who were newly invited to join BreastScreen, aged 50-74 years, and most overdue for screening. In contrast, clients with least priority were existing clients who reached the target age range (50-74 years) and whose scheduled appointment fell during the recovery period.

The following text has been added to the manuscript, in the methods section, under the section entitled ‘Breast screening’:

“Following service resumption the restart strategy involved the allocation of available screens according to (i) whether clients were newly invited or existing, (ii) their age; (iii) the period in which their appointment fell (i.e. during the pause or during the recovery period), and (iv) the time elapsed since their originally scheduled appointment. Appointments were prioritised for women in the target age range of 50-74 years and to clients most overdue for screening.

For bowel screening, individuals missing a screen during the pause would be screened at the next screening round, 2 years later. The following text has also been added to the methods section, under ‘Bowel screening’ as follows:

“It was assumed that individuals missing a screen during the pause would be screened at the next screening round, 2 years later.”

For cervical screening, as noted in the Methods section, we assumed that women who missed screening in 2020, were instead screened in 2021-2022. To further clarify, we have adapted the text in the methods section, under ‘Cervical screening’ by adding the text:

“We explored two recovery scenarios, one where women who missed screening in 2020 all attended in 2021 (used with the 9-month disruption scenario), and another where women who missed screening in 2020 attended gradually over 2021-2022 (used with 6- and 12-month disruption scenarios)”.

The existing footnote in Table 1 has also been adjusted to further clarify by adding the text:

“Scenarios assumes women who miss screening in 2020, instead attend over 2021-2022"

2. In line 109 you mention: "60-84% of clients screened on schedule (first to third and subsequent round)". What is meant by this?

Thank you for identifying that this is a confusing statement. National BreastScreen data is usually reported according to screening round as follows: first round, second round, third round and subsequent rounds (>3+ rounds). The statement means that 60-84% of women participating in the national screening program, attended their scheduled appointment, whether that appointment corresponded to their 1^st^ -3^rd^ screen, or a screen following that. The sentence has been amended as follows:

“Prior to COVID-19, participation by women aged 50-74 was approximately 55%, with 51% of national breast cancers detected through BreastScreen, while the proportion of clients who attended screening on schedule ranged from 60% to 84%, depending on the screening round.”

3. In line 178 you mention a "cervical screening test (CST)". Do you mean a HPV-test?

Yes, CST is used in Australia to refer to an HPV test but for improved clarity for an international audience, we have replaced CST/ Cervical Screening Test throughout the article with HPV test.

Results:1. Table 3 shows the reduction in the number of screen-detected invasive cancers per 100,000 women. I am quite surprised to see that the number of screen-detected tumors after a disruption of 12 months only decreased by 24%, when there was no screening for an entire year and the screening program restarted gradually over the next 6 months. Is there a catch-up in those last 6 months when the normal screening program was again in place?

We thank the reviewer for bringing this to our attention. We assumed higher throughput following a 12-month disruption (150% of status quo) than following shorter disruptions (status quo throughput). This was not described previously. We have now amended the methods section of manuscript, under ‘Breast screening’ as follows:

“We assumed a gradual restoration of service following pause periods, returning by the seventh month to pre-COVID throughput (screening capacity) for shorter-term pauses (3- and 6 -months) and 150% of pre-COVID throughput for a longer pause (12 months), as a year-long pause would likely involve expanded recovery strategies.”

We have amended the relevant text in Table 1 for the Breast screening program as indicated by highlighted text below:

12 months: 12-month pause assuming gradual recovery over 6 months, to 50% higher screening capacity than status-quo

6 months: 6 month-pause assuming gradual recovery over 6 months, to status-quo screening capacity

3 months: 3-month pause assuming gradual recovery over 6 months, to status-quo screening capacity

In the Results section, under ‘Breast cancer screening’ to make this distinction in throughput between scenarios, we added the text as follows:

For the two-year period 2020-2021, we estimate that screening pauses of any duration would lead to an overall reduction in population-level invasive breast cancer diagnosis for women aged 50-74 years, noting that the estimated reduction in cancer incidence was comparable (~9%) between a 6-month pause with pre-COVID throughout during recovery and a 12-month pause with 150% throughput during recovery.

Similarly, the following additional text was added to the first paragraph of the discussion:

For a 6-month screening disruption we estimated reduced diagnoses of breast cancer (9.4% population-level reduction over 2020-2021, with 100% throughput) and colorectal cancer (up to 5.8% reduction over 2020-21) and increased cervical cancer diagnoses (up to 1.1% over 2020-2022), with less marked upstaging compared to a 12-month disruption.

2. Table 3 shows the reduction in the number of interval-detected invasive cancers. Does this concern the interval-detected invasive cancers of women screened in 2020/2021? Or does it concern the number of interval-detected invasive cancers diagnosed in 2020-2021?

The number of interval-detected invasive cancers in Table 3 refers to cancers diagnosed in 2020-2021. Footnote c, under Table 3, has now been amended as shown below:

c) Breast screening: invasive breast cancers arising within 12 or 27 months of a negative screening episode. Figures reflect interval cancers diagnosed in 2020, or 2020-2021, respectively.”

3. In line 259-260 you write that an estimated 759-3022 people would be affected by delays in diagnosis for bowel cancer. Where can I find those numbers?

These numbers are the combination of the delayed and upstaged diagnoses, though it differs from the sum of these to avoid double counting those affected by both a delay and an upstage. We have removed this for clarity as we do not feel this output is useful and may be confusing for this manuscript.

Discussion:1. I was quite surprised to see some figures in the Discussion. I would place figure 1 in the Results section. Also it would be very interesting to add an analysis showing the expected effect of what really happened with the screening program. This would also distinguish your paper from the other two papers I mentioned above.

Figure 1 draws on national data that became publicly available in 2021, well after our evaluation was conducted. Since this figure is not based on results from our study, and to prevent confusion, Figure 1 was placed in the Discussion. Although uncommon, in this particular case we believe the figure should remain in place to enable a better understanding of the timeline the different data became available. We are planning a detailed comparison of modelled and observed outcomes as more observed data becomes available. While this would be interesting, our paper is sufficiently unique.

2. I would place figure 2 in the appendix, since you already have Table 2.

Figure 2 has now been removed from the main manuscript and converted into supplementary figure 8. (Please note during final editorial review supplementary figure 8 was placed back into the main manuscript, as Figure 2).

3. Could you please explain your reasoning in line 376-378? I am not sure if I agree with the statement that your model-estimated reductions in screening tests in the first year of disruption overlapped with observed rates. Table 2 show that in Q1-Q3 still 82% of the tests were performed for breast cancer, while with a 3 months disruption only 53% of the test will be performed in April 2020-March 2021. I do not think those numbers overlap. The numbers are also not very comparable as the observed numbers are presented for Jan-Sept 2020 and the modelled numbers for April 2020-March 2021. It would be nice if you can take the same periods.

We agree that this statement is not sufficiently supported, and have removed it from the discussion. We will consider generating data for matching periods in a future more detailed comparison of observed and simulated data.

4. In line 413-414 you mention that your findings illustrate that maintaining screening participation is critical to reducing the burden of cancer at a population level. Could you please explain your reasoning? I believe that the numbers show that screening can safely be suspended for a few months if there are limited resources available. Only if you suspended the program for a year, problems may occur.

This comment is similar to comment 4, which has been addressed above. We have now amended the relevant sentence in the discussion as follows:

“Collectively, findings from the 6-12 month disruption scenarios illustrate that maintaining screening participation is critical to preventing an increase in the burden of cancer at a population level.”

Reviewer #3 (Recommendations for the authors):While the presented results by assessing various disrupted screening schedules (12 months, 9 months, 6 months, and 3 months) through statistical modelling are very helpful for guiding policy-makers to avert long-term impact on cause-specific mortality and cost for example in the post-pandemic COVID-19 era, the manuscript can be substantially improved in the light of the following suggestions.1. It is very important to report the influences of COVID-19 pandemic on long-term mortality resulting from the disrupted screening schedule although the reported outcomes here have already included the reduction of screen-detected cancers and the assessment of upstaging. The author has already indicated the unfeasibility of reporting mortality estimates because of unavailable information on COVID-19 disruptions to cancer treatment. However, since your modelling has assumed many situations on various delayed screening schedule it would not be difficult to model mortality estimates by setting various circumstances on COVID-19 disruptions to cancer treatment.

We agree with the reviewer that estimated mortality is important, and while this was not within the scope of the rapid-response modelling reporting here, these have now been published for some of the scenarios reported here for bowel and cervical screening, as part of cross-country modelling exercises [de Jonge et al., 2021; Smith et al., 2021]. We have added the following sentence to the methods section under ‘Reported Outcomes’.

Mortality estimates are not reported, however, estimates for some of the scenarios have since been published separately for cervical and bowel cancer screening^12,14^

2. While modelling screening episodes by various disrupted screening schedules plays an important role in policy analysis it is even important to report the empirical results of the delayed screening schedule until 31 March 2021 instead of only reporting the observed number from January to September 2020. You can still report your simulated results for policy analysis but report which scenario, on average, is close to your empirical estimate. So doing is very helpful for predicting the best-case outcomes after 2021.

We have included extensive discussion of the subsequent real-world decisions and developments in the relation to all three screening programs as part of the Discussion section. To address the reviewer’s point, we have now added the following text in the discussion under the section entitled ‘Subsequent observed outcomes’ to indicate that the most conservative modelled scenarios were closest to observed data:

‘For the year 2020, observed reductions in participation were lower than the most conservative estimates modelled for each program, with 14% fewer breast screening tests^26^, an estimated 7% reduction in bowel screening participation,^26^ and for cervical screening, based on reimbursement claims data, a 10% reduction in expected HPV tests.^27-28’^

3. It is quite unreasonable to have seen reduced interval cancers and the reduced program sensitivity with the disruption scenario from the aspect of screening theory. If disruption to screening happen those cancers would be delayed to have diagnosis. It is therefore inadequate to define interval cancer in this manner in the face of disruption scenario, saying an artefact of reduced screening participation. The reader is very confused with such a kind of results. As interval cancer is often a negative outcome in the theory of screening, the results of reducing interval cancer as a result of disruption to screening would be very misleading. The longer disruption the less likely to have interval cancer. This part should be rectified to have adequate presentation.

We agree that this can be confusing, but it would not be appropriate for us to introduce an alternative definition of interval cancers, particularly given that BreastScreen did not change its definitions. We do explain this complication clearly in the second paragraph of the discussion (‘noting that estimated reductions in population-level interval cancer rates are counter-intuitive but an artefact of reduced screening participation, as interval cancers are defined with reference to screening events').

4. Instead of reporting absolute number of upstaging with different expressions for various cancers, it is indispensable to report relative risk of having upstaging for various disruption scenarios as opposed to status quo by breast cancer, colorectal cancer, and cervical cancer base on the modelling results. So doing this can be also of assistance to evaluate statistical significance by reporting 95% confidence interval and also P value for hypothesis testing.

As these are modelled results rather than observed outcomes, we would not apply the measures of error suggested. Therefore, we have not made the suggested change.

5. The results of upstaging are so important for this manuscript particularly when mortality estimates have been not reported. However, the modelling results on upstaging seems to vary with disruption scenarios. This part has to be emphasized in the main conclusions for different cancer. Again, using the relative risk estimates would make the comparisons of the finding across three sites of cancer comparable.

We have now provided additional information on upstaging of cancers for all disruption scenarios for bowel and cervical cancer screening in Table 3. However, we have not added relative risk estimates as these metrics would differ between cancer types; due to differences in model structures, the breast and bowel models report the proportion of high-grade cancers, and cervical cancer reports the proportion of cancers that were upstaged.

6. The point 2 mentioned above is also important for validating the simulated results of screen-detected cases, upstaging and other outcomes with the scenario in commensuration with the empirical scenario of disruption to screening.

We agree that further validation of simulated and observed outcomes would be of value, and this will be the subject of a future evaluations.